# Validity and reliability of an app-based medical device to empower individuals in evaluating their physical capacities

Alexandre Mazéas [1,2,3]*, Marine Blond[3], Aïna Chalabaev[1], Martine Duclos[2,4]

**1** SENS, Univ. Grenoble Alpes, Grenoble, France, **2** INRAE, UNH, CRNH Auvergne, Clermont Auvergne University, Clermont-Ferrand, France, **3** Kiplin, Nantes, France, **4** Department of Sport Medicine and Functional Exploration, University Hospital Clermont-Ferrand, Hospital G. Montpied, Clermont-Ferrand, France

* alexandre.mazeas@univ-grenoble-alpes.fr

## Abstract

### Background

Cardiorespiratory fitness and muscle strength are valid markers of health and strong predictors of mortality and morbidity. The tests used to measure these variables require in-person visits with specialized equipment and trained personnel–leading to organizational constraints both for patients and hospitals, and making them difficult to implement at a large scale. In this context, technologies embedded in smartphones offer new opportunities to develop remote tests.

### Objectives

This study aimed to test the validity and reliability of *MediEval*, a newly developed app-based medical device that allows individuals to perform the 6-minute walk test (6MWT) and the 30-second sit-to-stand (30s-STS) test on their own using GPS signal and camera detection with a skeleton extraction algorithm.

### Methods

A total of 53 healthy adults performed the two tests in three different sessions to determine the intra- and inter-day reproducibility. Test validity was assessed by comparing the results obtained from the app to gold standard measures. Pearson correlations and concordance correlation coefficients, the relative measurement error, intraclass correlation coefficients, the standard error of measure and the minimal detectable change were computed for each test.s

### Results

The results revealed high to excellent validity of the app in comparison to gold standards ($\rho$ = 0.84 for the 6MWT and $\rho$ = 0.99 for the 30s-STS test) with low relative measurement error. The mean differences between the app and the gold standard measures were 8.96m for the 6MWT and 0.28 repetition for the 30s-STS test. Both tests had good test-retest

**Data Availability Statement:** The anonymized data used in this study and the R code are available on the Open Science Framework (https://osf.io/4hzke/).

**Funding:** The work of AM is supported by an ANRT grant (Cifre PhD Thesis) and by the company Kiplin. The funders had no role in study design, data collection and analysis, decision to publish, or preparation of the manuscript.

**Competing interests:** The authors of this manuscript have read the journal's policy and have the following competing interests: AM's PhD grant is funded by the French National Association for Research and Technology (ANRT) and Kiplin. MB is employed by Kiplin. This does not alter our adherence to PLOS ONE policies on sharing data and materials.

reliability (ICCs = 0.77). The minimal detectable changes were respectively 97.56 meters for the 6MWT and 7.37 repetitions for the 30s-STS test.

## Conclusion

The *MediEval* medical device proposes valid and reproducible measures of the 6MWT and the 30s-STS test. This device holds promise for monitoring the physical activity of large epidemiologic cohorts while refining patient experience and improving the scalability of the healthcare system. Considering minimal detectable change values, it may be important to ask participants to perform several tests and average them to improve accuracy. Future studies in clinical context are needed to evaluate the responsiveness and the smallest detectable changes of the device for specific populations with chronic diseases.

## Introduction

Cardiorespiratory fitness and muscle strength are important predictors of the overall health of individuals–being strongly associated with reduced mortality, reduced risk of developing chronic diseases, and improvement in the functional capacities and autonomy [1–5]. The evaluation of these two dimensions is essential as physical activity is part of patients' non-drug therapy [6]. This evaluation allows monitoring the functional capacities at the level of an individual or a population, tailoring supervised physical activity programs, measuring the effectiveness of these programs, informing the patients on their health, and ultimately empowering them.

The 6-minute walk test (6MWT) is a commonly used self-paced test for the objective assessment of functional exercise and cardiorespiratory capacity. This test measures the distance patients walk on a flat and hard surface over a six-minute period [7]. The 6MWT is used in the general adult population, but also in older adults, or subjects with chronic conditions such as type 2 diabetes, osteoarthritis, cardiopulmonary disease, stroke, or Parkinson's disease [8]. Healthy subjects generally cover a distance of around 682 (±73) meters in men and 643 (±70) meters in women [9], while in patients, performance is generally less important and more variable depending on the pathology. The reliability, validity, and responsiveness of the 6MWT have been extensively tested and validated across various clinical settings [8, 10, 11]. The popularity of the 6MWT can be explained by its many assets. This submaximal test is safer, easier to administer, better tolerated, and better reflects daily life activities than high-intensity or incremental tests [12, 13]. Moreover, it is simple to analyze and interpret, inexpensive, and takes less than 10 minutes to conduct.

The chair sit-to-stand (STS) test involves the functional movement of rising from a seated position and is frequently used to assess lower-limb muscular strength [14]. Several variations of the STS test have been described in the literature [8] including the 30s-STS test, which measures the number of stands achieved in 30 seconds. This test is used for a wide range of populations including hip and knee osteoarthritis or young and older adults [8]. Normative scores for the 30s-STS test in community-dwelling older people are around 14.2 repetitions (±4.6) among men and 12.7 repetitions (±4.0) among women [15] and around 33 repetitions (±5.4) among healthy young populations [16]. This test has acceptable test-retest reliability (for a review see [8]) and moderate-to-high correlations with lower limb strength [17, 18]. In addition, this test is often used to assess the functional capacity of older adults to predict and

prevent falls [19]. Similar to the 6MWT, the 30-second STS is easy to administer, analyze, and interpret. It requires little equipment, can be performed in any environment, and takes no more than 3 minutes to complete [8, 20, 21].

Despite their many apparent benefits, these tests involve, in practice, high costs and limitations for both patients and healthcare institutions. On the one hand, these tests entail travel costs and additional stress for patients, who have to come specifically to the hospital/platform to perform the tests. On the other hand, implementing these tests leads to organizational constraints for the healthcare center that must set up a dedicated corridor, and involve significant human resource costs with the presence of a specialized professional to conduct the tests, analyze, and interpret the results. More recently, the in-person requirement for these tests also made them difficult to implement during the COVID-19 pandemic. For all these reasons, these two tests cannot be performed on a regular basis whereas they could be conducted in the absence of trained professionals if the patient has the appropriate tools to easily and accurately measure performance [12].

To address these challenges, m-health tools represent a promising perspective. With the widespread availability of affordable smartphones and internet access (14 billion mobile devices and 4.9 billion internet users in the world in 2021 [22]) the majority of the population now owns a smartphone, which enables the remote completion of the tests near or at the patient's home. As early as 2011, Wevers et al. [23] demonstrated that the 6MWT could be performed outdoors using a global positioning system (GPS) or a measuring wheel (reproducible, responsive, and valid test), suggesting that this test could be performed from patients' homes. Matthew et al. [24] suggested the feasibility of using a single-depth camera to assess STS movements, opening new assessment perspectives. In addition, the feasibility of remote 30s-STS tests appears good as a recent study suggested that a video-guided STS test is suitable for participants of varying ages body sizes, and activity levels [25]. In consequence, it would be conceivable to estimate the distance walked or the number of repetitions performed during the tests using the smartphone's GPS or smartphone camera of the user.

In this perspective, the Kiplin company developed in collaboration with the present authors and the CEA tech Nantes a medical device to empower patients in conducting the 6MWT and the 30s-STS test. *MediEval* is a stand-alone software module integrated within a mobile app and certified class 1 medical device under the European Medical Device Regulation. This device offers a new opportunity to monitor individuals' physical health status and symptoms over time due to the faculty of performing more tests independently [12].

However, in order to use this device with confidence in clinical settings, information regarding the potential error and precautions in using this app need to be investigated in real-world conditions. The aims of the present study are to test the validity and reliability of this app-based medical device to evaluate the cardiorespiratory fitness and lower limb muscle strength of healthy individuals in a natural environment. Based on previous research and preliminary testing of the app, we hypothesized that measurements conducted with the *MediEval* device would be highly correlated with gold standards measurements, and that the device would provide reproducible measurements (inter- and intra-subject).

## Methods

### System design/ app development

*MediEval* is a class 1 medical device (Unique Device Identifier 3770024180008) allowing individuals to perform the 6MWT and the 30s-STS test, in autonomy. This app module is incorporated within the Kiplin app (available on iOS and Android smartphones, with iOS version 13

and Android version 7 as minimum configurations). Subjects need to have a physical activity prescription to access the content of *MediEval*.

**Technical functioning of the 6MWT.** The distance traveled by the user during the 6MWT is computed on the basis of the phone's location by GPS position, recorded every 5 seconds [26]. The triangulation of the GPS points is then used to calculate the distance traveled and the associated speed in a straight line over each 5-second interval. The total distance walked is computed by summing the distances obtained for each interval. The distance measured on the outlier intervals (i.e., intervals where anormal speed is detected), is then corrected on the basis of our algorithm.

**Technical functioning of the 30s-STS test.** The number of STS movements performed by the user is determined by an algorithm applied to the video stream transmitted by the phone. The first step is to apply a skeleton extraction carried out by a state-of-the-art algorithm [27] on each image obtained through the video stream, which provides angle values for the whole skeleton and allows analysis of the user's biomechanical movements. Specific angle values are then used to classify the user's posture using a binary decision tree. Likewise, a binary decision tree is used to qualify the detected posture as correct or incorrect (especially to check the position of the arms, which need to be crossed at the wrists and held against the chest). The number of correct STS sequences is used to calculate the test result.

**User experience.** Through written and video tutorials, as well as the validation of a checklist before each test (Fig 1B), the user is invited to respect the following instructions:

- the 6MWT must be performed outdoors, on a flat surface, with no curves and no risk of GPS obstruction (no tall buildings or trees), where the user will be able to go back and forth for 100 meters;

- the STS test must be performed in a bright room, using a traditional chair (i.e., at about knee height when standing ≈ 43cm; as specified in the instructions) without armrests, and the user needs to position his phone on another chair at a sufficient distance so that he can be filmed by the phone's camera.

For the 6MWT, the user cannot start the test if the GPS accuracy of his phone does not stabilize under 15m for at least 5 seconds as the start button is not available. If the accuracy is acceptable, a clickable Play button allows the user to start the test, and a timer displaying the time remaining until the end appears (Fig 1D). The app also emits a vibration every minute to indicate the remaining time. The end of the 6-minute test is indicated to the user on the screen and by means of vibrations of the phone. A long vibration and a display on the screen indicate the end of the test.

For the STS test, in order to ensure that the camera has been positioned properly, the user is invited to take a photo to check that the camera captures the user's body from head to toe when he or she is standing (Fig 1C). Then, a clickable Play button allows starting the test with a 15-second countdown for the user to take place. During the test, the time remaining over is displayed on the screen, with a beep sounding for each STS movement detected as correct. Similarly, when an incorrect STS sequence is detected, a different beep sounds. The end of the test is indicated to the user on the screen and by a long beep.

At the end of the tests, the user is asked to evaluate his level of muscular fatigue and dyspnea using Borg's scale. The app then displays the result of the test, including the distance achieved in meters and the comparison with the theoretical distance that the user should perform (i.e., calculated according to Enright and Sherrill's equations [28] for the 6MWT, and the number of correct repetitions (i.e., 30s-STS test score), and incorrect movements detected for the STS test (Fig 1E).

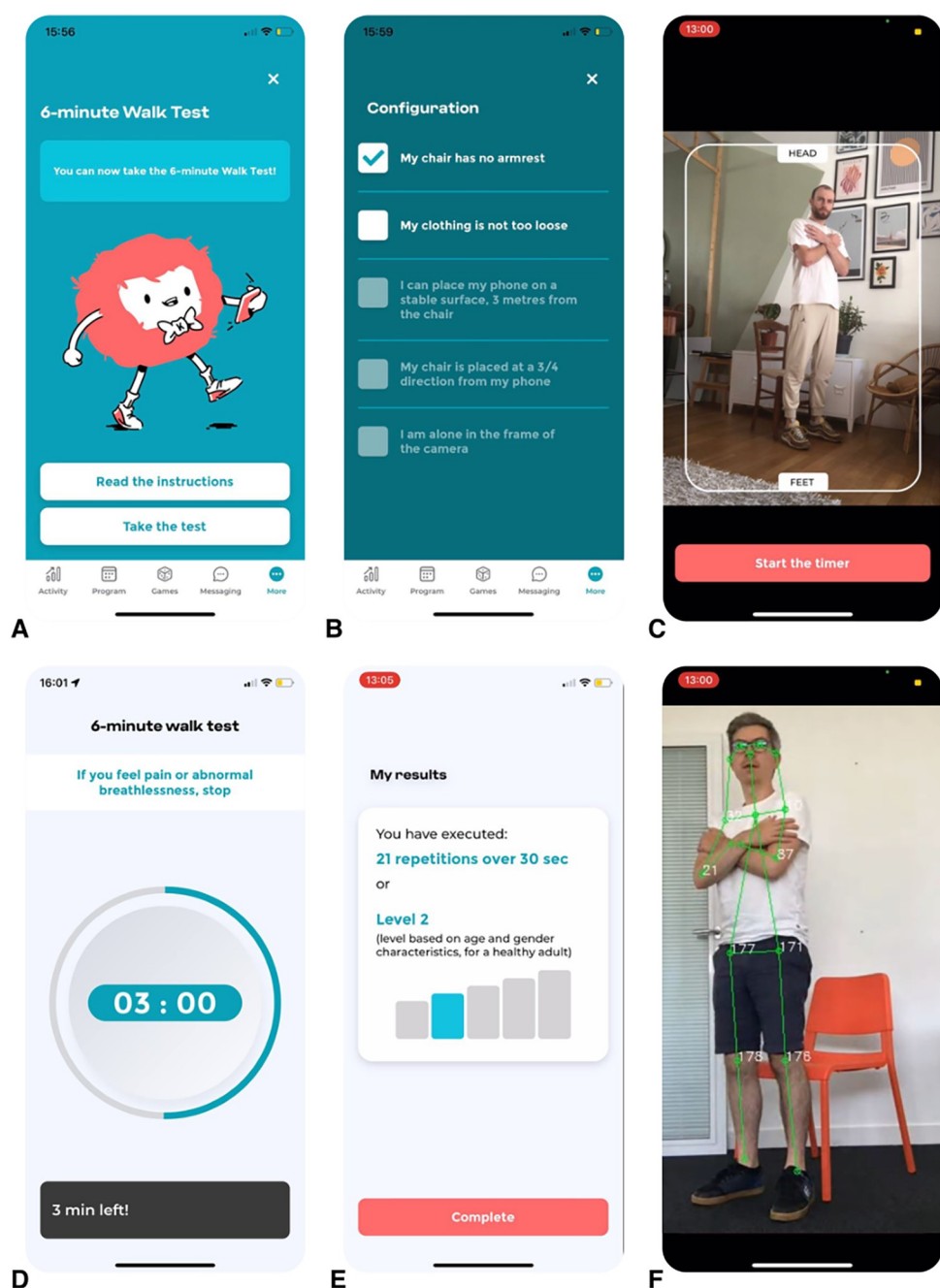

**Fig 1. Screenshots of the *MediEval* medical device.** (A) Welcome screen for the 6MWT. (B) Checklist for the 30s-STS test. (C) Photo taken to check that the camera captures the user's body from head to toe for the 30s-STS. (D) Screen during the 6MWT. (E) Results for the 30s-STS test. (F) Detection of the skeleton algorithm (this view is not available for the user). Reprinted under a CC BY license, with permission from the Kiplin company, original copyright [2023].

Usability tests with participants of various ages conducted prior to this study did not raise any problems of ergonomics, usability, or comprehension.

## Participants

A total of 53 healthy adults aged 21 to 63 years old (mean age = 33 ± 10.9 years; Body Mass Index (BMI) = 22.8 ± 3.0 kg/m$^2$; 57% women) volunteered to participate in the current study.

Participants must be aged >18 years and answer 'no' to all items of the Physical Activity Readiness Questionnaire for Everyone (PAR-Q+, [29]) to be included. They were excluded in case of injury, surgery, or any disease pathology which could affect their physical aptitude or their gait. Participants were enrolled between May and October 2022. The study protocol was approved by the local ethics committee (IRB00012476-2022-26-04-177). Written informed consent was obtained from all participants and all analyzed information was obtained exclusively from anonymous data.

## Procedure and measures

*Criterion validity* refers to the extent to which scores obtained from the app are related to a gold standard [30]. For the 6MWT, we compared the distance computed by the app to the distance measured via a distance wheel accurate to 0,1 m (M20, GEO FENNEL, Germany), which is a commonly accepted gold standard of distance measurement [23, 31, 32]. For the STS test, the video stream generated during the test was recorded and analyzed a posteriori by two observers instructed to the guidelines of the STS. A third observer was consulted in case of discrepancy between the first observers. This video analysis was considered the gold standard [25] and compared to the app's result. Retrospective visual analysis via video recordings is a common gold standard measure in physical activity and condition validation studies (e.g., [33–35]).

*Reproducibility* concerns the degree to which repeated measurements in stable persons during a test-retest procedure provide similar answers [30]. In this context, the time period between the repeated administrations should be long enough to ensure recuperation, though short enough to ensure that clinical change has not occurred [30]. The test-retest should therefore not be conducted more than 2 weeks apart. Within-day reproducibility is also of interest to assess the potential effect of time of day or circadian cycles on test reliability. In the present study, each participant was invited to perform a 6MWT and a 30s-STS test during three independent sessions. The first two sessions were conducted on the same day in order to measure the intra-day reproducibility of the device. These sessions were conducted in the morning and in the afternoon, at a minimal 6-hour interval (maximum 12 hours), in order to ensure the participants' recovery. The last session was scheduled several days later (minimum 1 week; maximum 2 weeks) in the morning at the same time as the first session in order to evaluate the inter-day reproducibility of the tests.

*Interpretability* refers to the extent to which scores obtained from the app can be interpreted by providing reference data from the general population [36]. In other words, interpretability is capital in regard to change scores to be able to affirm if a change in the measured performance should be considered part of the measurement error or as a real change [30, 36]. Interpreting change in test scores implies two metrics: the measurement error, expressed as the minimal detectable change (MDC), and the minimal important change (MIC). On the one hand, the MDC reflects the smallest within-person change in score that can be interpreted as a "real" change, above measurement error. Thus, a change score can only be considered to represent a real change if it is larger than the MDC. On the other hand, the MIC represents the smallest measured change score that patients perceive to be important [37]. The MDC needs to be smaller than the MIC to precisely distinguish a clinically important change from measurement error [38].

**Full procedure.** In the first session, participants answered the PAR-Q+ and signed the consent form. Each participant performed the tests with their own phone in order to approach real-life measurements and to control the smartphone brand in the analyses. To do so, participants downloaded the Kiplin app on their phones and entered their demographic information

on *MediEval*. This included age, gender, height, and body weight to calculate BMI ($kg/m^2$). Previous research showed that adult self-report of weight and height is strongly correlated with objectively measured values [39]. Then, they watched the video tutorial explaining the tests.

After the experimenter has verified the correct understanding of the instructions, the participants performed the 6MWT on an athletics track. The test consisted of round trips on a 100m straight line, delimited by two marks (which the participants had to turn around). We chose such settings based on the preliminary testing of the device as this distance was a good compromise between optimal conditions for GPS recognition (since GPS points are measured every 5-second interval to calculate the distance traveled, too many round trips or a non-rectilinear trajectory can lead to a loss of data) and feasibility (asking participants to perform the test on an 800-meter straight line seem not feasible). At the end of the 6 minutes, the participant stopped and the total distance covered was measured with the distance wheel.

In a second time, participants had to perform the 30s-STS test, after having done a few warm-ups and set up their phones in the appropriate conditions. All participants performed the test on the same standard-size chair. The number of repetitions was noted by the experimenter and the video of the test was recorded for later verification.

The other two sessions followed the same procedure. Participants did not receive monetary compensation. For both tests, participants were asked to strive for the best performance.

## Statistical analyses

**Sample size and power analysis.** We conducted an a priori sample size estimation based on 1) preliminary results of the device in internal tests and the available scientific literature that allowed us to expect a reliability (intraclass correlation coefficient, ICC) of 0.85 ($\rho1$), and 2) the recommendations of Terwee et al. [30], who proposed a minimal acceptable reliability (ICC) of 0.70 ($\rho0$). This power analysis revealed that 53 participants were needed to reach 80% power and a two-sided type I error at 0.05.

**Data analysis.** Criterion validity was assessed by calculating the Person's correlation coefficient (the normality of the distribution was checked with a Shapiro-Wilk test) between the scores given by the app and the score measured via the gold standard. The scoring system for correlation coefficients as described by McCall [40] was used: 0.0–0.2 very low or negligible; 0.2–0.4 low; 0.4–0.7 moderate, 0.7–0.9 high; and > 0.9 very high. Validity was considered convincing when the correlation with the gold standard was at least 0.70 [30]. The difference between the app-based and gold standard-based measures was transformed into the relative measurement error (*rME*), which provides the ratio of the absolute error to the measurement in comparison to the gold standard. Concurrent validity of the app and the gold standard scores was assessed via the Lin's concordance correlation coefficient (CCC). Regular cut-off values of CCC coefficients can be considered as follow: < 0.70 very poor; 0.70–0.90 poor; 0.90–0.95 moderate; 0.95–0.99 good [41]. Systematic differences between the two measures were investigated with Bland & Altman plots [42]. In addition, the linear regressions between both methods were plotted.

Reproducibility can be divided into two different constructs: the *reliability* concerns the degree to which individuals can be distinguished from each other despite measurement error [43] whereas the *agreement* concerns the absolute measurement error (i.e., how close the scores on repeated measures are). Two-way random effects ICC (2,1) was used to assess reliability, as it is the most appropriate and commonly used reliability parameter for continuous measurements [30]. Usually, 0.70 is recommended as the minimum standard for reliability [44]. Between-person and within-period variances were estimated with a linear mixed effects

model for absolute agreement, adjusted for age, BMI, and type of smartphone. The agreement was computed as the standard error of measurement (SEM) [45] which represents the standard deviation of repeated measures in one patient, and was calculated from the square root of the error variance of the ICC ($\sqrt{\text{VarError}}$). The coefficient of variation (calculated as standard deviation / mean × 100) was also computed for both tests. Bland-Altman plots were performed to visualize agreement.

Finally, for interpretability, the MDC was calculated as $1.96 \times \sqrt{2} \times \text{SEM}$. Sensitivity analyses were carried out on the basis of gender and age, with comparisons between men and women and between <30 and ≥30 years old (we dichotomized in this order to obtain two groups of similar size; 30 years being the median age of our sample). All analyses were conducted using R (R Foundation for Statistical Computing). The data and code for the statistical analyses used in the present study are available on Open Science Framework (https://osf.io/4hzke/).

## Results

A total of 158 measurements were obtained for each test. Participants' demographics are reported in Table 1.

### Criterion validity

Results indicated a high correlation between the 6MWT distance measured by the *Medieval* app and the distance measured via the distance wheel ($\rho = 0.84$, $p < 0.001$) and a very high correlation between the STS test scores obtained with the app and observed scores ($\rho = 0.99$, $p < 0.001$). Fig 2 shows Bland-Altman plots comparing the app scores with the gold standard measure for both tests. For the 6MWT, the plot illustrates a mean difference of 8.96 meters and a 95% limit of agreement of −76.96 to 94.88 meters whereas the plot relative to the STS test reveals a mean difference of -0.28 repetition and a 95% limit of agreement of −2.16 to 1.70 repetitions. The linear regressions between the Medieval and gold standard measures are plotted in Fig 3. Examination of the plots suggests the existence of outliers for both tests. The mean |rME| for the 6MWT was 4.40% and 1.73% for the STS test. The CCC coefficients revealed respectively poor (0.84) and good (0.99) concurrent validity of the app and the score measured via the gold standard for the 6MWT and the STS test.

**Table 1. Descriptive statistics.**

| Descriptive statistics | |
|---|---|
| *Demographics* | |
| Age years, mean (SD) | 33.18 (11.00) |
| Female/male | 30/23 |
| BMI kg/m$^2$, mean (SD) | 22.64 (2.84) |
| *Type of smartphone used* | |
| iPhone (% of all tests) | 24 (44%) |
| Google Pixel (% of all tests) | 8 (14%) |
| Samsung Galaxy (% of all tests) | 14 (26%) |
| Huawei (% of all tests) | 3 (6%) |
| Xiaomi (% of all tests) | 3 (6%) |
| OnePlus (% of all tests) | 1 (2%) |
| Realme (% of all tests) | 1 (2%) |
| *Test scores* | |
| 6MWT meters, mean (SD) | 704.55 (79.13) |
| 30se-STS repetitions, mean (SD) | 21.87 (5.89) |

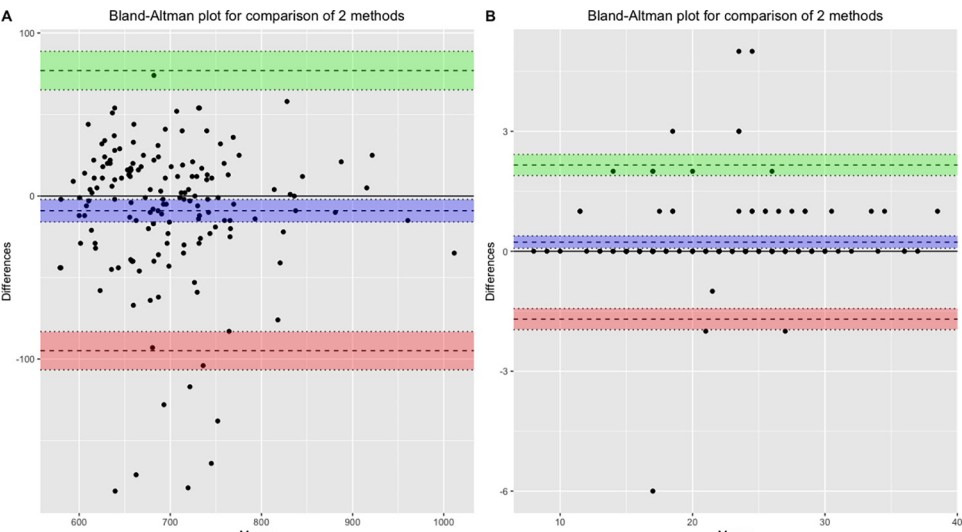

**Fig 2. Bland-Altman plots for validity.** (A) Comparison of the app-based and distance wheel measure of the 6MWT distance. (B) Comparison of the app-based and observed performance at the 30s-STS test. The upper and lower lines represent the 95% limits of agreement, and the center line indicates the mean difference.

For the 6MWT, mean |rME| stratified by phone brands ranged as the following: OnePlus (1.14%), Huawei (2.22%), Realme (2.44%), iPhone (3.14%), Samsung Galaxy (4.67%), Google Pixel (5.06%), and Xiaomi (20.1%). The |rME| of Xiaomi phones was significantly higher than the other brands ($p < 0.001$) and corresponds to the outliers visible on the regressions plots (see Fig 4 for the linear regressions stratified by type of smartphone used for the tests). Thus, when excluding Xiaomi phones in sensitivity analyses, the correlation between the scores app and the distance measured via the distance wheel was $\rho = 0.89$, with a mean |rME| of 3.78%.

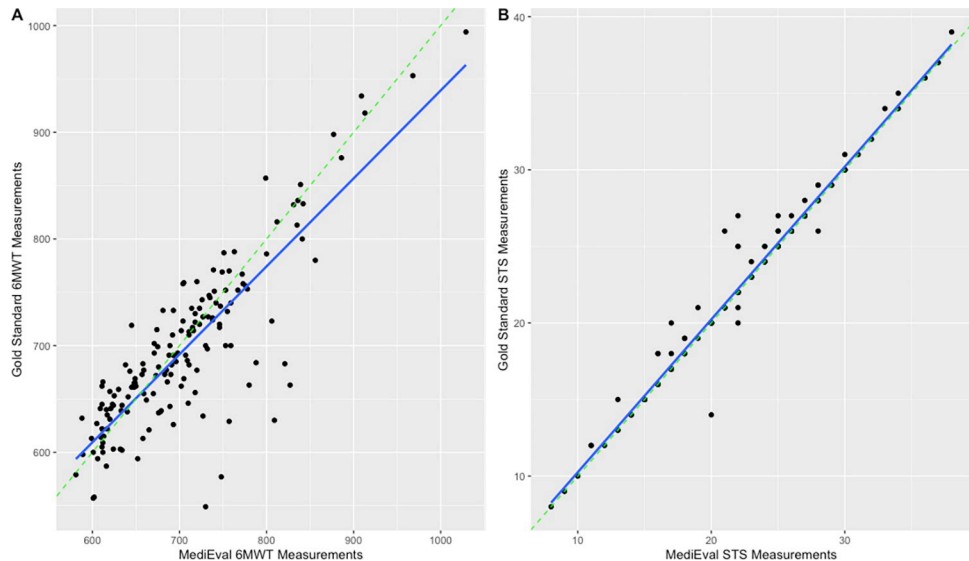

**Fig 3. Linear regressions plots for validity.** (A) Comparison of the app-based and distance wheel measure of the 6MWT distance. (B) Comparison of the app-based and observed performance at the 30s-STS test. The dotted green line at 45 degrees represents equality between the results.

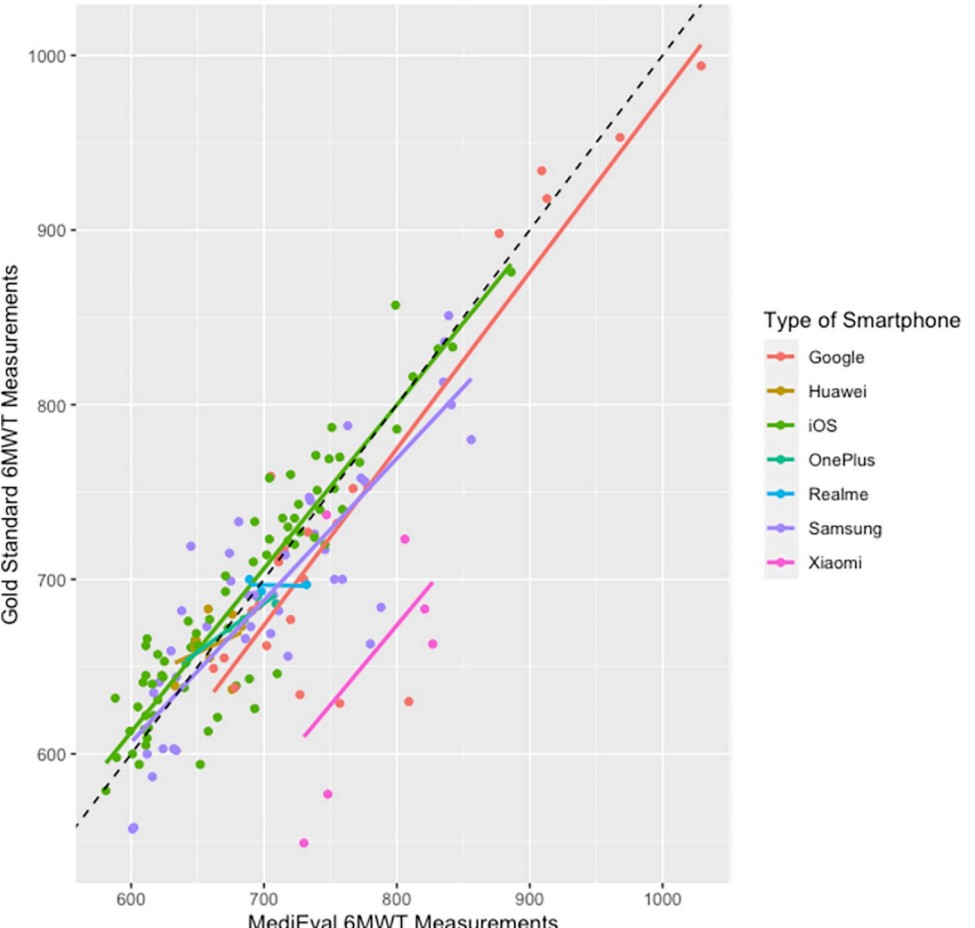

**Fig 4. Linear regression plot for validity between the app-based and distance wheel measure of the 6MWT distance with stratification by type of smartphone.** The dotted black line at 45 degrees represents equality between the results.

### Reproducibility and interpretability

Test-retest reliability estimates of the *MediEval* indicated high intra-day stability for the 6MWT ($ICC_{2,1} = 0.83$) and the STS test ($ICC_{2,1} = 0.79$) and respectively high and moderate inter-day reproducibility for the 6MWT ($ICC_{2,1} = 0.72$) and the STS test ($ICC_{2,1} = 0.68$). The adjusted ICCs considering the three measurement points were 0.67 for 6MWT and 0.70 for the STS test. ICC values were higher without adjustment for age, BMI, and type of smartphone ($ICC_{2,1} = 0.78$ for both tests). As a comparison, the $ICC_{2,1}$ for the gold standard measures of the 6MWT and the STS test were respectively $ICC_{2,1} = 0.78$ and $ICC_{2,1} = 0.75$. The coefficient of variation was 9.96% for the 6MWT and 25.84% for the STS test. The SEM for the 6MWT was 35.20 meters and 2.66 repetitions for the STS test, which provided an MDC of 97.56 meters (13.85%) for the 6MWT and 7.37 repetitions (33.72%) for the STS test. Finally, Bland-Altman plots comparing the test-retest app scores for both tests reveal intra-day mean differences of respectively -7.19 meters (95% limit agreement [-19.01; 4.63]) for the 6MWT and -1.32 repetition (95% limit agreement [-2.30; -0.34]) for the STS test. Inter-day mean differences were -27.35 meters (95% limit agreement [-42.72; -11.97]) for the 6MWT and -2.04 repetitions (95% limit agreement [-3.25; -0.83]) for the STS test (Fig 5). A summary of the findings is available in Table 2.

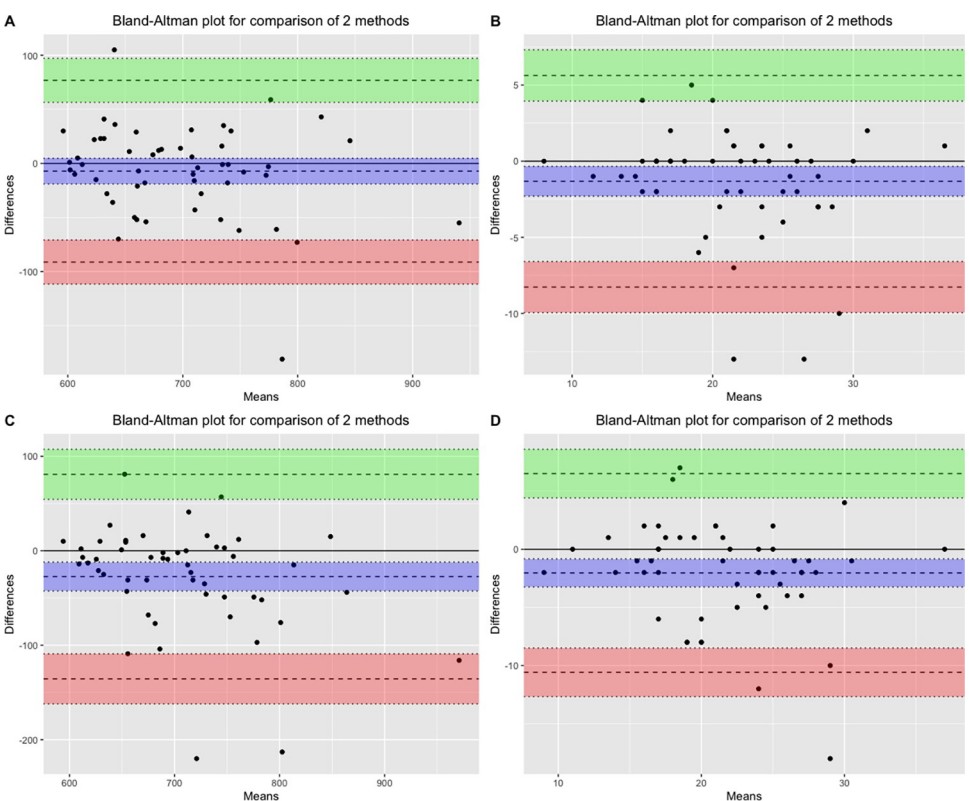

**Fig 5. Bland-Altman plots for reproducibility.** (A) Intra-day test-retest for the 6MWT. (B) Intra-day test-retest for the 30s-STS test. (C) Inter-day test-retest for the 6MWT. (D) Inter-day test-retest for the 30s-STS test. The upper and lower lines represent the 95% limits of agreement, and the center line indicates the mean difference.

## Sensitivity analyses

Sensitivity analyses comparing men and women results and "young" and "older" adults are available in supplementary materials (S1 and S2 Tables). If the CCC is lower in women than men (0.73 vs 0.90), the reliability is better among women on both tests ($ICC_{2,1} = 0.83$ vs 0.75 on the 6MWT and $ICC_{2,1} = 0.89$ vs 0.58 on the 30s-STS test), the SEM lower (26.77 vs 43.24 meters for the 6MWT and 3.10 vs 2.16 repetitions for the 30s-STS test) and in consequence the MDC also lower (74.20 vs 119.74 meters for the 6MWT and 8.59 vs 5.99 repetitions for the

**Table 2. Summary of the results.**

| Variables | 6MWT (95%CI) | 30s-STS test (95%CI) |
|---|---|---|
| Validity (Pearson correlation) | 0.84 (0.80; 0.88) | 0.99 (0.98; 0.99) |
| Relative measurement error (%) | 4.40 | 1.73 |
| Concordance correlation coefficient (CCC) | 0.84 (0.78; 0.88) | 0.99 (0.98; 0.99) |
| Reliability ($ICC_{2,1}$) | 0.78 (0.67; 0.86) | 0.78 (0.66; 0.86) |
| Intra-day reliability ($ICC_{2,1}$) | 0.83 (0.73; 0.90) | 0.79 (0.65; 0.88) |
| Inter-day reliability ($ICC_{2,1}$) | 0.72 (0.49; 0.84) | 0.68 (0.46; 0.82) |
| Standard error measurement (meters and repetitions) | 35.20 (28.11; 42.28) | 2.66 (2.66; 3.18) |
| Coefficient of variation (%) | 9.96 | 25.84 |
| Minimal detectable change (meters and repetitions) | 97.56 | 7.37 |
| Minimal detectable change (%) | 13.85 | 33.72 |

30s-STS test). Regarding the age stratification, results from participants $\geq 30$ years old showed better correlation with gold standard ($\rho = 0.86$ vs $0.78$ and CCC = $0.86$ vs $0.77$) and higher reliability (ICC$_{2,1}$ = $0.83$ vs $0.75$) than younger participants for the 6MWT.

## Discussion

### Principal findings

The main contribution of the current study was to develop and evaluate the validity and reproducibility of an app-based medical device aimed to empower individuals in conducting physical condition tests on their own. The results revealed high to excellent validity of the app in comparison to gold standards–with high correlations and low rME–which suggests that the *MediEval* device was precisely able to measure the physical performance. Indeed, the mean differences between the app and the gold standard measures were 8.96 meters for the 6MWT and 0.28 repetition for the 30s-STS test.

Moreover, the reliability and agreement of the device during the test-retest were good and similar to the reliability of scores obtained with gold standard measures. However, regarding interpretability, the SEM and MDC of both tests were high, with a relative MDC of 13.85% (absolute MDC = 97.56 meters) for the 6MWT and 33.72% (absolute MDC = 7.37 repetitions) for the 30s-STS test. Sensitivity analyses suggested that reliability, SEM and MDC values were better among women than men whereas the validity and reliability of the 6MWT measure was higher among 'older' adults.

### Comparison with previous research

Our results for the 6MWT are comparable with previous studies that showed good acceptability, validity, and reliability of devices that used the GPS coordinates of the smartphone to calculate the 6MWT distance outdoors [26, 32]. For the 30s-STS test, this study is the first to design and test the innovative way of assessing STS movements by camera detection using a skeleton extraction algorithm. Other options for physical functioning remote assessment are to conduct the tests via videoconferencing technology [46] or using video-guided self-administered tests [25]. If these two options also showed positive validity estimates, these methods require personnel to administer or evaluate the test where the present device allows patients to perform the test independently.

Regarding interpretability, previous research that estimated the MIC for the two tests concluded that a minimum change in the 6MWT distance of 45 meters was considered to be clinically meaningful [47] while an increase greater than or equal to 2.6 repetitions on the 30s-STS test can be associated with a major improvement [48]. The MDCs found in this study were slightly larger than these values. Therefore, if an individual has a change score as large as the MIC but lower than the MDC, we cannot be 95% sure that this change is not due to measurement error. In other words, individual change scores calculated with the *MediEval* medical device that would be below the MDC values should be interpreted with caution.

Nevertheless, considering the low rME observed and the fact that reliability measures were identical between the app and gold standard (ICC$_{2,1}$ = $0.77$ for the app and ICC$_{2,1}$ = $0.78$ and $0.75$ for the gold standard), we can assume that these high MDCs could be attributable to irregularities in the performance of the participants recruited in the present study–who performed differently at the different experimental sessions. Often known as the learning effect, the 6MWT distance tends to increase during the first 5 tests [49]. Moreover, the performance realized by healthy individuals can be more heterogenous than patients with limited physical capacities. Since a) the sources of error are maximized in the case of high performance, due to the fact that more distance can be covered between two GPS points and more movements can

be made between two video analysis frames by the detection algorithm, and b) the gap between two performances will be reduced in the case of lower initial performance, we can imagine that the validity and reproducibility of the device will be better in patients or fragile people who will achieve lower performances on the two tests. The results of our sensitivity analyses suggesting better reproducibility among older people and women support this idea. Future tests of *MediEval* on clinical contexts are needed to properly determine the MDC of each test for specific populations.

## Strengths and limitations

The major strength of this study is the validation of an innovative medical device in real-life conditions among a population of various ages who used their own smartphone to perform the tests. Indeed, these tests have been conducted in everyday-life reproducible conditions, whereas previous studies evaluating GPS-based 6MWT assessment apps have been conducted in a lab environment and settings that do not seem practically feasible for the patient (e.g., performing the 6MWT on a straight course of 500–700m). The generalization of the *Medieval* medical device on a large scale will also be facilitated by the fact that the app is available on both iOS and Android smartphones (covering the vast majority of the global mobile operating system market share with cumulatively >99% [50]).

Despite these positive assets, it is important to stress that this study was conducted on a small, healthy sample, which reduces the generalizability of our results beyond the scope of this population. Second, the 6MWT that needs to be performed outdoors to get an accurate GPS signal can still be complicated to implement for patients and several factors can limit the accuracy of the GPS such as atmospheric fluctuations, ephemeris error, satellite clock drift, hardware error, and unfavorable satellite geometry that may lead to incorrect measurement of distance [51]. Third, as demonstrated by the SEM and MDC analyses, using a single test to interpret the effect of a program with relatively small change scores may not be enough to get an accurate measurement. Finally, in this study participants followed the instructions of the examiners to perform the tests. We can question whether the tests will be performed correctly in a real-life setting when the patients are alone.

This study also revealed that a GPS-based remote 6MWT is not as accurate on all smartphones. The statistical analyses showed that Xiaomi smartphones had a significantly higher rME than the other brands tested. This could be explained by a poorer-quality GPS embedded in these phones. Large longitudinal data collection in the post-market surveillance process will allow adapting the algorithms according to the GPS accuracy and the type of smartphone. In the meantime, the results of 6MWT tests performed with Xiaomi smartphones should be interpreted with caution.

## Perspectives for future research and implications for practice

The first step in developing and evaluating *MediEval* was to ensure that the device would validly and reproducibly measure performance on fitness tests under supervised conditions in healthy individuals. The next step is to test this medical device in clinical settings among patients with chronic diseases. Such study will be important to determine the responsiveness of the device (i.e., the ability to detect clinically important changes over time). Since this device could generate important money savings and facilitate the onboarding of patients in physical activity programs, it will also be essential to evaluate the economic impact of the use of a such device for the healthcare system. As mentioned earlier, it will also be capital in future studies to test that the remote tests are performed properly in accordance with the instructions given in the tutorials with the collection of real-world data.

From a practical point of view, one way of decreasing the MDC is to conduct and average multiple measurements (i.e., repeated measurements at one point in time) in order to decrease the measurement error [36]. Thus, it could be asked to participants using *MediEval* to perform at least two tests in a short period to obtain a more accurate measure and control for the learning effect. Moreover, the present results suggesting good validity and reliability of the device, *MediEval* opens new perspectives for measuring the strength, mobility, and physical function of large epidemiologic cohorts. Such device could therefore be a complement to physical activity measurements performed on representative samples of the general population at the country level–allowing ultimately to determine and monitor the physical condition of the populations of a country. As physical capacities are important markers of health both in young and in adults, healthy or with chronic diseases, the development of a cost-effective measure of physical capacities that could be implemented in medical offices, in hospital settings but also at home can help to determine subjects at high risks [52].

## Conclusions

Traditional physical fitness and muscular strength tests require in-person visits with specialized equipment and trained personnel, leading to organizational constraints both for patients and hospitals and making them difficult to implement on a large scale. *MediEval*, an app-based medical device, allows participants to conduct the 6MWT and the 30s-STS test remotely and in autonomy. The present validating study revealed that this device conveniently measures participants' performances with good validity and reproducibility estimates on healthy participants. However, this study showed that, when taking into account MDC and MIC, the change score should exceed 97 meters for the 6MWT and 7.37 repetitions for the 30s-STS test to consider a clinically relevant change which is not due to measurement error. Averaging multiple measurements could be a way to reduce these values. Future studies will evaluate the responsiveness and validity of this device in clinical settings among patients with chronic diseases and provide specific MDC values for each population.

## Supporting information

**S1 Table. Summary of the results stratified by age (participants $<$ 30 years old vs participants $\geq$ 30 years old).**
(PDF)

**S2 Table. Summary of the results stratified by sex (men vs women).**
(PDF)

## Acknowledgments

The authors want to thank Charlotte Herbrecht (research assistant) for helping out with the measurements and all the participants for their time and involvement.

## Author Contributions

**Conceptualization:** Alexandre Mazéas, Marine Blond, Martine Duclos.

**Data curation:** Alexandre Mazéas.

**Formal analysis:** Alexandre Mazéas.

**Investigation:** Alexandre Mazéas.

**Methodology:** Alexandre Mazéas.

**Software:** Alexandre Mazéas.

**Writing – original draft:** Alexandre Mazéas.

**Writing – review & editing:** Marine Blond, Aïna Chalabaev, Martine Duclos.

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
