## [Decision Letter · Decision Letter 0]

31 May 2023

PONE-D-23-10193Validity and reliability of an app-based medical device to empower patients in evaluating their physical capacitiesPLOS ONE

Dear Dr. Mazeas,

Thank you for submitting your manuscript to PLOS ONE. After careful consideration, we feel that it has merit but does not fully meet PLOS ONE’s publication criteria as it currently stands. Therefore, we invite you to submit a revised version of the manuscript that addresses the points raised during the review process.

We look forward to receiving your revised manuscript.

Kind regards,

Duncan S Buchan

Academic Editor

PLOS ONE

“have read the journal's policy and the authors of this manuscript have the following competing interests: AM’s PhD grant is funded by the French National Association for Research and Technology (ANRT) and Kiplin. MB is employed by Kiplin.”

5. We note that Figure 1 in your submission contain copyrighted images. All PLOS content is published under the Creative Commons Attribution License (CC BY 4.0), which means that the manuscript, images, and Supporting Information files will be freely available online, and any third party is permitted to access, download, copy, distribute, and use these materials in any way, even commercially, with proper attribution. For more information, see our copyright guidelines: http://journals.plos.org/plosone/s/licenses-and-copyright.

b.If you are unable to obtain permission from the original copyright holder to publish these figures under the CC BY 4.0 license or if the copyright holder’s requirements are incompatible with the CC BY 4.0 license, please either i) remove the figure or ii) supply a replacement figure that complies with the CC BY 4.0 license. Please check copyright information on all replacement figures and update the figure caption with source information. If applicable, please specify in the figure caption text when a figure is similar but not identical to the original image and is therefore for illustrative purposes only.

Reviewers' comments:

Reviewer's Responses to Questions

**Comments to the Author**

1. Is the manuscript technically sound, and do the data support the conclusions?

Reviewer #1: Yes

Reviewer #2: Partly

2. Has the statistical analysis been performed appropriately and rigorously? 

Reviewer #1: No

Reviewer #2: Yes

3. Have the authors made all data underlying the findings in their manuscript fully available?

Reviewer #1: Yes

Reviewer #2: Yes

4. Is the manuscript presented in an intelligible fashion and written in standard English?

Reviewer #1: Yes

Reviewer #2: Yes

5. Review Comments to the Author

Reviewer #1: 1- General comment

I congratulate the authors for their time and effort in conducting this study, with very interesting findings for clinical and research settings. Indeed, using an app to collect data is always a significant advance in our area because it is cost-effective, time- and space-efficient, easy to use, and accessible to anyone. However, although the article presents important content for the scientific community due to the relevant practical application in clinical, sports, and research settings, the main limitation is the small sample size (although the authors justify it) and the inclusion of only healthy adults, which makes these results not generalizable to other sample populations. In addition, the statistical analysis can be improved with additional tests. Below, I provide some suggestions.

2- Abstract

- It should be updated based on the results obtained by the analysis recommended below.

3- Introduction

- The introduction section is well-written and presents the rationale and the research problem. The readers can easily understand the need to develop the current study. I suggest adding the study hypothesis.

- LL51-52: “This test measures the distance patients walk on a flat and hard surface over a six-minute period.” Please, add references.

- LL57-58: “The chair sit-to-stand (STS) test involves the functional movement of rising from a seated position and is frequently used to assess lower-limb muscular strength.” Please, add references where the different variations are included.

4- Methods

- LL121: How was the height of the chair controlled? Did all participants comply with the value? This factor plays an important role in the STS test.

- LL164-165: One must be very careful when interpreting the results, particularly when the comparison is made with a gold standard, recently elaborated, where further studies are needed to confirm this assumption (STS test).

- LL167-169: “In this context, the time between the repeated administrations should be long enough to ensure recuperation, though short enough to ensure that clinical change has not occurred.” Please, add references.

- LL199: Data analysis:

- What was the test used for the correlation analysis? Why? Were the assumptions met? Please, indicate it.

- Besides correlation coefficients (Spearman’s rank?), please present the linear regression between both methods. Add a line of 45 degrees in the regression graph to demonstrate the equality between results. These are all measures of concurrent validity.

- Please, indicate the ICC model and the ICC interpretation. You can follow this article DOI: 10.1016/j.jcm.2016.02.012. In addition, please present the 95% confidence intervals for the ICC.

- In addition to the ICC analysis, please add the Lin’s concordance correlation coefficient (CCC) to assess the concurrent validity of the app and the score measured via the gold standard. For interpreting the CCC, please see DOI: 10.1002/uog.13320.

-In addition to the SEM, please present the coefficient of variation (%) and interpret it adequately.

- I recommend that the authors calculate the minimal detectable change (MDC) and present absolute and relative values. Please, see 10.1016/j.pmrj.2017.05.001 to support the calculations.

-LL186: “100m straight line”. I believe this distance is too high when comparing the usual distances used in clinical settings (15-30 m). Why did the authors choose 100 m instead of the usual distance? Is it related to smartphone detection constraints? Please, report it in the manuscript.

5- Results

The results subsection should be updated with the recommended analyses.

- Table 2: Please, present the 95% CI.

6- Discussion

- This subsection should be updated based on the results provided by the recommended analyses.

- LL281: Comparison with previous research. This subsection needs to be improved. Although the literature might be scarce, comparing the current results with those observed with different methods is important. In addition, by including the MDC analysis, the authors can compare their results with those found for the 6MWT and the 30CST.

- LL291: Please comment on the lack of generalization of the current results due to the small sample size and the health status (only healthy adults).

Reviewer #2: The study aims at evaluating the reliability and validity of a novel software/mobile app MediEval. The manuscript is quite well written, and readability is also excellent. The statistical analyses are detailed and apparently appropriate. The discussion section too focuses on the strength, limitation and other stuff but it does not discuss much about the reliability and SEM things.

From fitness assessment instrument points of view, the manuscript is on spot. However, this novel software/app is supposed mostly for patients or clinical purposes and therefore, some contents seem to be contradicted or can be potentially misleading. Please address these issues carefully before publication as I suppose clinicians, or some patients may potentially use this software / apps for medical purposes such as progress monitoring and making decision of subsequent treatment/rehabilitation plans. The requirement of instrument accuracy, information regarding the potential error and precautions in using this app need to be more rigorous and well informed.

1) In the introduction, please provide a bit more general and clinical applications in 6MWT and STS tests, as well as their approximate distances / scores. E.g. COPD? Hypertension? Cardiac surgery? If it can be also be used for non-clinical condition, is it also a common physical fitness test for normal adults or elderly? Now all your subjects are healthy individuals but if 6MWT and STS is not commonly used by normal adults or these tests do not provide lots of meaningful values in general fitness, the study design using healthy subjects does not match the focus of introduction and discussion section and it will become unsound as well.

2) The major weakness or limitation of the paper is that it only includes healthy subjects and most score/distance range obtained in this study (e.g. 704.55m for 6MWT) is far different from real patients. Therefore if majority of contents in the introduction or discussion are to promote this software/app for clinical use, the obtained results cannot be highly generalized to clinical populations. For example, the SEM value of 6MWT is 35.2m but quite a lot of patients with chronic disease could only walk for about 150-250m. The minimal clinically important difference in traditional 6MWT test can be just 30m as a cutoff point (PMID: 26252533). If simply reading your SEM values without careful understanding and analysis, it sounds like such a large SEM value makes this app/software unusable as the clinician or patient will have no idea if the observed changes of +30m is simply measurement error or reflecting a true improvement. Therefore, you should mention these potential discrepancies in your discussion. Moreover, from your Bland-Altman plot, although not very explicit, it seems that the observed difference becomes larger when the means score are getting larger (e.g. 750-900 vs. 600-750). Therefore, the SEM value may be substantially smaller when this test is applied on fragile individuals.

3) I prefer to also reporting the minimum/smallest detectable changes to tell the 95% confidence interval instead of just SEM values. However, the MDC value can be 97.5 m in 6MWT. Even though the author claimed good test-retest reliability (ICCs = 0.77), when it is used in a real life situation, it is very difficult to have a patient or healthy individual to achieve >97.5 m improvement in the post-test. If so, I doubt the test may not be sensitive enough to detect the real change in real life condition. Of course, when measuring the SEM or MDC values in fragile groups with substantially slower walking speed, the value should be much lower than 35.2 (SEM) or 97.5 (MDC) values. I think the author should also address these issues in the discussion instead of just saying the device reliable (interestingly, the author has lightly mentioned this in the last sentence of the abstract but nothing in the discussion part)

4) What are the instructions to subjects in performing 6MWT and STS? Walking as fast as they can? Natural pace?...

5) Why were there some outliers? The author tried referring all these outliers to Xiaomi. Can you adjust those plots to the color by phone group such that we can visualize if ALL outliers only come from Xiaomi but nothing else? This open access journal allow colorful visualization and I think col by groups can be read easily. If outliers exist in other conditions but not related to Xiaomi, while this product in the future is to be used for medical purposes, the author may need to explain a bit more why or when the large discrepancy exists....as you don't want to put extra risk on patients due to even the occasional but massive error induced in unknown situations.

6) Line 304 - lead to an overestimation.....however from your Bland-Altman plot, it seems underestimation is little bit more common. I think the author should address more on proposing the explanation of those observed discrepancies. Meanwhile if possible, compare the MDC or SEM values to previous similar studies. I suggest that the authors may consider adding the ICC, SEM and MDC value based on different subject groups (age in 2 groups if not enough subjects for each sub-group? gender or phone device)

The authors have given some long paragraphs regarding the potential implications and applications in the discussion section while discussion on the potential error, accuracy issues, reliability values are much shorter and a bit lacking. It may give the impression to readers that the authors lean on selling the apps rather than keeping the discussion in neutral. I think most of the discussion is to explain the findings and compare them with previous research as well as proposing reasonable explanations on them. The practical application of findings is good to make research findings more usable in the real world. While most potential advantages should already been provided in the introduction section to justify why doing this study for this apps. Otherwise, the content may not perfectly match your big topic "validity and reliability".

6. PLOS authors have the option to publish the peer review history of their article (what does this mean?). If published, this will include your full peer review and any attached files.

Reviewer #1: No

Reviewer #2: **Yes: **Indy Man Kit Ho

---

## [Author Response · Author response to Decision Letter 0]

7 Jul 2023

Rebuttal letter: Point-by-point response to the editor and reviewers’ comments.

Dear Dr. Duncan S Buchan,

We thank you for providing us the opportunity to revise this manuscript. Many thanks also to the different reviewers for their queries, recommendations, and suggestions, all of which were much appreciated and very helpful in strengthening the manuscript.

Please find below our point-by-point responses to the editor’s and reviewers’ comments. Comments are in bold font, our responses are in regular font, quotes from the manuscript are in brackets, and each change made in the manuscript is in blue font. 

Best regards,

The authors

RESPONSE: We updated the name of our files following the PLOS ONE style requirements. 

RESPONSE: As indicated in our manuscript, all R statical coeds and anonymized data are available on the Open Science Framework. 

“have read the journal's policy and the authors of this manuscript have the following competing interests: AM’s PhD grant is funded by the French National Association for Research and Technology (ANRT) and Kiplin. MB is employed by Kiplin.”

RESPONSE: We updated the competing interests section following your recommendation. 

CHANGE: “The authors of this manuscript have read the journal's policy and have the following competing interests: AM’s PhD grant is funded by the French National Association for Research and Technology (ANRT) and Kiplin. MB is employed by Kiplin. This does not alter our adherence to PLOS ONE policies on sharing data and materials.” (Lines 531-534, declaration of interests section). 

RESPONSE: We removed the ethics statement at the end of our manuscript. 

5. We note that Figure 1 in your submission contain copyrighted images. All PLOS content is published under the Creative Commons Attribution License (CC BY 4.0), which means that the manuscript, images, and Supporting Information files will be freely available online, and any third party is permitted to access, download, copy, distribute, and use these materials in any way, even commercially, with proper attribution. For more information, see our copyright guidelines: http://journals.plos.org/plosone/s/licenses-and-copyright.

b.If you are unable to obtain permission from the original copyright holder to publish these figures under the CC BY 4.0 license or if the copyright holder’s requirements are incompatible with the CC BY 4.0 license, please either i) remove the figure or ii) supply a replacement figure that complies with the CC BY 4.0 license. Please check copyright information on all replacement figures and update the figure caption with source information. If applicable, please specify in the figure caption text when a figure is similar but not identical to the original image and is therefore for illustrative purposes only.

RESPONSE: We have the permission to publish this from the copyright holder (and also permission from the people in the photo). In consequence, we uploaded the completed content permission form and updated the legend of Figure 1. 

Reviewer #1: 1- General comment

I congratulate the authors for their time and effort in conducting this study, with very interesting findings for clinical and research settings. Indeed, using an app to collect data is always a significant advance in our area because it is cost-effective, time- and space-efficient, easy to use, and accessible to anyone. However, although the article presents important content for the scientific community due to the relevant practical application in clinical, sports, and research settings, the main limitation is the small sample size (although the authors justify it) and the inclusion of only healthy adults, which makes these results not generalizable to other sample populations. In addition, the statistical analysis can be improved with additional tests. Below, I provide some suggestions.

RESPONSE: Many thanks for your precious comments and suggestions.

2- Abstract

- It should be updated based on the results obtained by the analysis recommended below.

RESPONSE: We updated the abstract with the new analyses and associated discussion. 

3- Introduction

- The introduction section is well-written and presents the rationale and the research problem. The readers can easily understand the need to develop the current study. I suggest adding the study hypothesis.

RESPONSE: Thank you for your comment. We added the study hypothesis at the end of the introduction section. 

CHANGE: We added the sentence: “Based on previous research and preliminary testing of the app, we hypothesized that measurements conducted with the MediEval device would be highly correlated with gold standards measurements, and that the device would provide reproducible measurements (inter- and intra-subject).” (Lines 121-123). 

- LL51-52: “This test measures the distance patients walk on a flat and hard surface over a six-minute period.” Please, add references.

RESPONSE: We added the reference of the ATS statement specifying the guidelines for the 6MWT. 

- LL57-58: “The chair sit-to-stand (STS) test involves the functional movement of rising from a seated position and is frequently used to assess lower-limb muscular strength.” Please, add references where the different variations are included.

RESPONSE: We added at the end of this sentence the reference of Bennell et al. (2011) which specifies the original development and different variations of the chair sit-to-stand test. 

4- Methods

- LL121: How was the height of the chair controlled? Did all participants comply with the value? This factor plays an important role in the STS test. 

RESPONSE: We added a sentence to specify that all participants performed the tests with the same chair in our study. 

CHANGE: We added the sentence: “All participants performed the test on the same standard-size chair.” (Lines 239-240). 

- LL164-165: One must be very careful when interpreting the results, particularly when the comparison is made with a gold standard, recently elaborated, where further studies are needed to confirm this assumption (STS test).

RESPONSE: The use of video analysis to count the number of STS repetitions closely reflect reality, and is also a gold standard used in many validation studies in sports sciences. We have added a sentence to support this argument.

CHANGE: We added the sentence: “Retrospective visual analysis via video recordings is a common gold standard measure in physical activity and condition validation studies (e.g., [33–35]).” (Lines 197-199). 

- LL167-169: “In this context, the time between the repeated administrations should be long enough to ensure recuperation, though short enough to ensure that clinical change has not occurred.” Please, add references.

RESPONSE: We added the reference of Terwee et al. (2007). 

- LL199: Data analysis:

- What was the test used for the correlation analysis? Why? Were the assumptions met? Please, indicate it.

RESPONSE: We performed Person correlations after having checked for the normality (with a Shapiro Wilk test) and linearity (by visual inspection) of our data. We have specified the type of test in the data analysis section of the method part and in Table 2.

CHANGES: We updated the sentence: “Criterion validity was assessed by calculating the Person’s correlation coefficient (the normality of the distribution was checked with a Shapiro-Wilk test) …” (Lines 252-253) and added “Validity (Pearson correlation)” in Table 2. 

- Besides correlation coefficients (Spearman’s rank?), please present the linear regression between both methods. Add a line of 45 degrees in the regression graph to demonstrate the equality between results. These are all measures of concurrent validity.

RESPONSE: We added a linear regression graph with a 45 degrees line for each test (Figure 3). 

- Please, indicate the ICC model and the ICC interpretation. You can follow this article DOI: 10.1016/j.jcm.2016.02.012. In addition, please present the 95% confidence intervals for the ICC.

RESPONSE: We specified the ICC model (i.e., ICC2,1) in the methods section, each time the term is specified in the result section, and in Table 2. We also added the 95% CI for the ICCs in Table 2. 

- In addition to the ICC analysis, please add the Lin’s concordance correlation coefficient (CCC) to assess the concurrent validity of the app and the score measured via the gold standard. For interpreting the CCC, please see DOI: 10.1002/uog.13320.

RESPONSE: We performed and presented the CCC scores for both tests. 

CHANGES: “Concurrent validity of the app and the gold standard scores was assessed via the Lin’s concordance correlation coefficient (CCC). Regular cut-off values of CCC coefficients can be considered as follow: < 0.70 very poor; 0.70–0.90 poor; 0.90–0.95 moderate; 0.95–0.99 good [41].” (Lines 260-263) and “The CCC coefficients revealed respectively poor (0.84) and good (0.99) concurrent validity of the app and the score measured via the gold standard for the 6MWT and the STS test.” (Lines 308-310). 

-In addition to the SEM, please present the coefficient of variation (%) and interpret it adequately.

RESPONSE: We performed and presented the coefficient of variation for both tests. 

CHANGES: “The coefficient of variation (calculated as standard deviation / mean × 100) was also computed for both tests.” (Lines 274-275) and “The coefficient of variation was 9.96% for the 6MWT and 25.84% for the STS test.” (Line 336). 

 - I recommend that the authors calculate the minimal detectable change (MDC) and present absolute and relative values. Please, see 10.1016/j.pmrj.2017.05.001 to support the calculations.

RESPONSE: We computed and presented absolute and relative MDC for both tests. In addition, we also added a paragraph on interpretability in the method section in order to emphasize the need to calculate MDC. 

CHANGES: “Interpretability refers to the extent to which scores obtained from the app can be interpreted by providing reference data from the general population [36]. In other words, interpretability is capital in regard to change scores to be able to affirm if a change in the measured performance should be considered part of the measurement error or as a real change [30,36]. Interpreting change in test scores implies two metrics: the measurement error, expressed as the minimal detectable change (MDC), and the minimal important change (MIC). On the one hand, the MDC reflects the smallest within-person change in score that can be interpreted as a ‘‘real’’ change, above measurement error. Thus, a change score can only be considered to represent a real change if it is larger than the MDC. On the other hand, the MIC represents the smallest measured change score that patients perceive to be important [37]. The MDC needs to be smaller than the MIC to can precisely distinguish a clinically important change from measurement error [38].” (Lines 213-224) and “Finally, for interpretability, the MDC was calculated as 1.96 × √2 × SEM.” (Lines 276-277) and “The SEM for the 6MWT was 35.20m and 2.66 repetitions for the STS test, which provided an MDC of 97.56 meters (13.85%) for the 6MWT and 7.37 repetitions (33.72%) for the STS test.” (Lines 337-338). 

-LL186: “100m straight line”. I believe this distance is too high when comparing the usual distances used in clinical settings (15-30 m). Why did the authors choose 100 m instead of the usual distance? Is it related to smartphone detection constraints? Please, report it in the manuscript.

RESPONSE: We conducted the tests on a 100-meter straight line, as we considered this to be the best compromise between the technological constraints of GPS recording and the feasibility and convenience of the user's experience. We've added a sentence to clarify these points.

CHANGE: “We choose such settings based on the preliminary testing of the device as this distance was a good compromise between optimal conditions for GPS recognition (since GPS points are measured every 5-second interval to calculate the distance traveled, too many round trips or a non-rectilinear trajectory can lead to a loss of data) and feasibility (asking participants to perform the test on an 800-meter straight line seem not feasible).” (Lines 234-238). 

5- Results

The results subsection should be updated with the recommended analyses.

- Table 2: Please, present the 95% CI.

RESPONSE: We updated the results section and Table 2 in line with your recommendations. 

6- Discussion

- This subsection should be updated based on the results provided by the recommended analyses.

RESPONSE: We incorporated and discussed the new results obtained with your recommended analyses. 

- LL281: Comparison with previous research. This subsection needs to be improved. Although the literature might be scarce, comparing the current results with those observed with different methods is important. In addition, by including the MDC analysis, the authors can compare their results with those found for the 6MWT and the 30CST.

RESPONSE: We compared the MDC obtained in our study to minimal important change of the two tests computed in the literature. 

CHANGE: “Regarding interpretability, previous research that estimated the MIC for the two tests concluded that a minimum change in the 6MWT distance of 45 meters was considered to be clinically meaningful [47] while an increase greater than or equal to 2.6 repetitions on the 30s-STS test can be associated with a major improvement [48]. The MDCs found in this study were slightly larger than these values. Therefore, if an individual has a change score as large as the MIC but lower than the SDC, we cannot be 95% sure that this change is not due to measurement error. In other words, individual change scores calculated with the MediEval medical device that would be below the MDC values should be interpreted with caution.” (Lines 395-402). 

- LL291: Please comment on the lack of generalization of the current results due to the small sample size and the health status (only healthy adults).

RESPONSE: We compared the MDC obtained in our study to minimal important change of the two tests computed in the literature. 

CHANGE: “Despite these positive assets, it is important to stress that this study was conducted on a small, healthy sample, which reduces the generalizability of our results.” (Lines 427-428). 

Reviewer #2: The study aims at evaluating the reliability and validity of a novel software/mobile app MediEval. The manuscript is quite well written, and readability is also excellent. The statistical analyses are detailed and apparently appropriate. The discussion section too focuses on the strength, limitation and other stuff but it does not discuss much about the reliability and SEM things.

RESPONSE: Thank you for your positive comments. 

From fitness assessment instrument points of view, the manuscript is on spot. However, this novel software/app is supposed mostly for patients or clinical purposes and therefore, some contents seem to be contradicted or can be potentially misleading. Please address these issues carefully before publication as I suppose clinicians, or some patients may potentially use this software / apps for medical purposes such as progress monitoring and making decision of subsequent treatment/rehabilitation plans. The requirement of instrument accuracy, information regarding the potential error and precautions in using this app need to be more rigorous and well informed.

RESPONSE: As this app-based medical device is embeded within the Kiplin app, patients cannot access it without a medical prescription and the supervision of a practitioner. This information is already specified in the methods part, “This app module is incorporated within the Kiplin app (available on iOS and Android smartphones, with iOS version 13 and Android version 7 as minimum configurations). Subjects need to have a physical activity prescription to access the content of MediEval.” (Lines 129-132). However, we agree that this is an important point and that some sections of our manuscript can be confusing. In consequence we added a sentence in introduction and changed the title. In accordance with your following points, we also nuanced a little more the discussion. 

CHANGES: “Validity and reliability of an app-based medical device to empower individuals in evaluating their physical capacities” (Title of the article) and “However, in order to use this device with confidence in clinical settings, information regarding the potential error and precautions in using this app need to be investigated in real-world conditions.” (Lines 118-119). 

1) In the introduction, please provide a bit more general and clinical applications in 6MWT and STS tests, as well as their approximate distances / scores. E.g. COPD? Hypertension? Cardiac surgery? If it can be also be used for non-clinical condition, is it also a common physical fitness test for normal adults or elderly? Now all your subjects are healthy individuals but if 6MWT and STS is not commonly used by normal adults or these tests do not provide lots of meaningful values in general fitness, the study design using healthy subjects does not match the focus of introduction and discussion section and it will become unsound as well.

RESPONSE: We added in the introduction information regarding the target population and normative scores for the 6MWT and the 30s-STS test. 

CHANGES: “The 6MWT is used in the general population, children, elderly, or chronic conditions such as osteoarthritis, cardiopulmonary disease, stroke, or Parkinson’s disease [8]. Healthy subjects generally cover a distance of around 682 (±73) meters in men and 643 (±70) meters in women [9], while in patients, performance is generally less important and more variable depending on the pathology.” (Lines 67-71) and “This test is used for a wide range of populations including hip and knee osteoarthritis, older adults, and children [8]. Normative scores for the 30s-STS test in community-dwelling older people are around 14.2 repetitions (±4.6) among men and 12.7 repetitions (±4.0) among women [15] and around 33 repetitions (±5.4) among healthy young populations [16].” (Lines 79-82). 

2) The major weakness or limitation of the paper is that it only includes healthy subjects and most score/distance range obtained in this study (e.g. 704.55m for 6MWT) is far different from real patients. Therefore if majority of contents in the introduction or discussion are to promote this software/app for clinical use, the obtained results cannot be highly generalized to clinical populations. For example, the SEM value of 6MWT is 35.2m but quite a lot of patients with chronic disease could only walk for about 150-250m. The minimal clinically important difference in traditional 6MWT test can be just 30m as a cutoff point (PMID: 26252533). If simply reading your SEM values without careful understanding and analysis, it sounds like such a large SEM value makes this app/software unusable as the clinician or patient will have no idea if the observed changes of +30m is simply measurement error or reflecting a true improvement. Therefore, you should mention these potential discrepancies in your discussion. Moreover, from your Bland-Altman plot, although not very explicit, it seems that the observed difference becomes larger when the means score are getting larger (e.g. 750-900 vs. 600-750). Therefore, the SEM value may be substantially smaller when this test is applied on fragile individuals.

RESPONSE: Thank you for this comment. We computed the minimal detectable change based on SEM values and compared it to the minimal important change observed in the literature for both tests. Then, we discussed the practical implications of such discrepancies between these values in the discussion. We also pointed out that these values could be smaller on more fragile individuals. 

CHANGES: “Regarding interpretability, previous research that estimated the MIC for the two tests concluded that a minimum change in the 6MWT distance of 45 meters was considered to be clinically meaningful [47] while an increase greater than or equal to 2.6 repetitions on the 30s-STS test can be associated with a major improvement [48]. The MDCs found in this study were slightly larger than these values. Therefore, if an individual has a change score as large as the MIC but lower than the SDC, we cannot be 95% sure that this change is not due to measurement error. In other words, individual change scores calculated with the MediEval medical device that would be below the MDC values should be interpreted with caution.” (Lines 395-402) and “Nevertheless, considering the low rME observed and the fact that reliability measures are identical between the app and gold standard (ICC2,1 = 0.77 for the app and ICC2,1 = 0.78 and 0.75 for the gold standard), we can assume that these high MDCs could be attributable to irregularities in the performance of the participants recruited in the present study – who performed differently at the different experimental sessions. Often known as the learning effect, the 6MWT distance tends to increase during the first 5 tests [49]. Moreover, the performance realized by healthy individuals can be more heterogenous than patients with limited physical capacities. Since a) the sources of error are maximized in the case of high performance, due to the fact that more distance can be covered between two GPS points and more movements can be made between two video analysis frames by the detection algorithm, and b) the gap between two performances will be reduced in the case of lower initial performance, we can imagine that the validity and reproducibility of the device will be better in patients or fragile people who will achieve lower performances on the two tests.” (Lines 403-414). 

3) I prefer to also reporting the minimum/smallest detectable changes to tell the 95% confidence interval instead of just SEM values. However, the MDC value can be 97.5 m in 6MWT. Even though the author claimed good test-retest reliability (ICCs = 0.77), when it is used in a real life situation, it is very difficult to have a patient or healthy individual to achieve >97.5 m improvement in the post-test. If so, I doubt the test may not be sensitive enough to detect the real change in real life condition. Of course, when measuring the SEM or MDC values in fragile groups with substantially slower walking speed, the value should be much lower than 35.2 (SEM) or 97.5 (MDC) values. I think the author should also address these issues in the discussion instead of just saying the device reliable (interestingly, the author has lightly mentioned this in the last sentence of the abstract but nothing in the discussion part)

RESPONSE: In line with your last comment, we computed MDC values. We also added the discrepancies between MDC and MIC in the strengths and limitation section. From a practical point of view we also suggested that conducting multiple measurements and averaging the score could be a strategy to improve reliability as suggested by previous research. 

CHANGES: “Third, as demonstrated by the SEM and MDC analyses, using a single test to interpret the effect of a program with relatively small change scores may not be enough to get an accurate measurement.” (Lines 432-434) and “From a practical point of view, one way of decreasing the MDC is to conduct and average multiple measurements (i.e., repeated measurements at one point in time) in order to decrease the measurement error [36]. Thus, it could be asked to participants using MediEval to perform at least two tests in a short period to obtain a more accurate measure and control for the learning effect.” (Lines 461-465). 

4) What are the instructions to subjects in performing 6MWT and STS? Walking as fast as they can? Natural pace?...

RESPONSE: Participants were asked to performed the best that they can. This information have been added in the methods section. 

CHANGE: “For both tests, participants were asked to strive for the best performance.” (Lines 243-244). 

5) Why were there some outliers? The author tried referring all these outliers to Xiaomi. Can you adjust those plots to the color by phone group such that we can visualize if ALL outliers only come from Xiaomi but nothing else? This open access journal allow colorful visualization and I think col by groups can be read easily. If outliers exist in other conditions but not related to Xiaomi, while this product in the future is to be used for medical purposes, the author may need to explain a bit more why or when the large discrepancy exists….as you don’t want to put extra risk on patients due to even the occasional but massive error induced in unknown situations.

RESPONSE: Thank you for this suggestion. We created a linear regression plot with a stratification based on the smartphone used during the test for the 6MWT (Figure 4). On this plot, we can clearly see that Xiaomi phones (the pink points) are outliers and tend to underestimate the measurements. 

6) Line 304 – lead to an overestimation.....however from your Bland-Altman plot, it seems underestimation is little bit more common. I think the author should address more on proposing the explanation of those observed discrepancies. Meanwhile if possible, compare the MDC or SEM values to previous similar studies. I suggest that the authors may consider adding the ICC, SEM and MDC value based on different subject groups (age in 2 groups if not enough subjects for each sub-group? Gender or phone device)

RESPONSE: Thank you for this observation and this suggestion. In consequence, we changed our sentence in the strengths and limitations section. Based on your recommendation, we also carried out sensitivity analyses in order to compare the results obtained on the basis of 2 important variables: sex (Male vs Female) and age. In order to compare two quite substantial samples, we dichotomized the variable with participants under and over 30 years of age. We chose this threshold because it was the median value of our sample and enabled us to obtain 2 equivalent groups. The results of this analyses are presented at the end of the results section and in supplementary materials. 

CHANGES: “several factors can limit the accuracy of the GPS such as atmospheric fluctuations, ephemeris error, satellite clock drift, hardware error, and unfavorable satellite geometry that may lead to incorrect measurement of distance [51].” (Lines 429-432) and “Sensitivity analyses were carried out on the basis of gender and age, with comparisons between men and women and between <30 and ≥30 years old (we dichotomized in this order to obtain two groups of similar size; 30 years being the median age of our sample).” (Lines 277-280). 

Lead to incorrect measurement

The authors have given some long paragraphs regarding the potential implications and applications in the discussion section while discussion on the potential error, accuracy issues, reliability values are much shorter and a bit lacking. It may give the impression to readers that the authors lean on selling the apps rather than keeping the discussion in neutral. I think most of the discussion is to explain the findings and compare them with previous research as well as proposing reasonable explanations on them. The practical application of findings is good to make research findings more usable in the real world. While most potential advantages should already been provided in the introduction section to justify why doing this study for this apps. Otherwise, the content may not perfectly match your big topic “validity and reliability”.

RESPONSE: Thank you for this comment. As suggested and in order to better meet the objective of our paper, and to take into account the new analyses carried out, we have nuanced the discussion and conclusion of our article a little more, pointing out the strengths, practical perspectives but also the limitations of this medical device.

---

## [Decision Letter · Decision Letter 1]

28 Jul 2023

Validity and reliability of an app-based medical device to empower individuals in evaluating their physical capacities

PONE-D-23-10193R1

Dear Dr. Mazeas,

We’re pleased to inform you that your manuscript has been judged scientifically suitable for publication and will be formally accepted for publication once it meets all outstanding technical requirements.

Kind regards,

Duncan S Buchan

Academic Editor

PLOS ONE

Additional Editor Comments (optional):

Reviewers' comments:

Reviewer's Responses to Questions

**Comments to the Author**

1. If the authors have adequately addressed your comments raised in a previous round of review and you feel that this manuscript is now acceptable for publication, you may indicate that here to bypass the “Comments to the Author” section, enter your conflict of interest statement in the “Confidential to Editor” section, and submit your "Accept" recommendation.

Reviewer #1: All comments have been addressed

Reviewer #2: All comments have been addressed

2. Is the manuscript technically sound, and do the data support the conclusions?

Reviewer #1: Yes

Reviewer #2: Yes

3. Has the statistical analysis been performed appropriately and rigorously? 

Reviewer #1: Yes

Reviewer #2: Yes

4. Have the authors made all data underlying the findings in their manuscript fully available?

Reviewer #1: Yes

Reviewer #2: Yes

5. Is the manuscript presented in an intelligible fashion and written in standard English?

Reviewer #1: Yes

Reviewer #2: Yes

6. Review Comments to the Author

Reviewer #1: Dear authors,

You did a great job in reviewing the manuscript as requested. I have no further comments. Congratulations on a very interesting study.

All the best

Reviewer #2: Congratulation to your great work and ready for publication. Thanks much for your detailed, rigorous and high-quality response. The manuscript is very well-written and excellent. Only 1 amendment is needed.

line 382: change SDC to MDC or you may say MDC (also known as SDC)

7. PLOS authors have the option to publish the peer review history of their article (what does this mean?). If published, this will include your full peer review and any attached files.

Reviewer #1: No

Reviewer #2: **Yes: **Indy Man Kit Ho

---

## [Editor Report · Acceptance letter]

2 Aug 2023

PONE-D-23-10193R1 

Validity and reliability of an app-based medical device to empower individuals in evaluating their physical capacities 

Dear Dr. Mazeas:

I'm pleased to inform you that your manuscript has been deemed suitable for publication in PLOS ONE. Congratulations! Your manuscript is now with our production department. 

Kind regards, 

on behalf of

Dr. Duncan S Buchan 

Academic Editor

PLOS ONE